# Resistin production does not affect outcomes in a mouse model of acute surgical sepsis

**Anthony S. Bonavia**[1,2]\*, **Zissis C. Chroneos**[3], **Victor Ruiz-Velasco**[1,2], **Charles H. Lang**[4,5]

**1** Department of Anesthesiology and Perioperative Medicine, Penn State Milton S Hershey Medical Center, Hershey, PA, United States of America, **2** Department of Pharmacology, Penn State College of Medicine, Hershey, PA, United States of America, **3** Department of Pediatrics, Penn State Milton S Hershey Medical Center, Hershey, PA, United States of America, **4** Department of Surgery, Penn State Milton S Hershey Medical Center, Hershey, PA, United States of America, **5** Department of Cellular and Molecular Physiology, Penn State College of Medicine, Hershey, PA, United States of America

\* abonavia@pennstatehealth.psu.edu

## Abstract

### Introduction

Because of the strong correlation between the blood concentration of circulating resistin and the illness severity of septic patients, resistin has been proposed as a mediator of sepsis pathophysiology. *In vitro* data indicate that human resistin directly impairs neutrophil migration and intracellular bacterial killing, although the significance of these findings *in vivo* remain unclear.

### Objective

The objectives of the present study were: (1) to validate the expression of human resistin in a clinically relevant, murine model of surgical sepsis, (2) to assess how sepsis-induced changes in resistin correlate with markers of infection and organ dysfunction, and (3) to investigate whether the expression of human resistin alters immune function or disease outcomes *in vivo*.

### Methods

107 male, C57BL/6 mice transgenic for the human resistin gene and its promoter elements ($Retn^{+/-/-}$, or Retn+) were generated on a $Retn^{-/-}$ (mouse resistin knockout, or Rko) background. Outcomes were compared between age-matched transgenic and knockout mice. Acute sepsis was defined as the initial 24 h following cecal ligation and puncture (CLP). Physiologic and laboratory parameters correlating to the human Sequential Organ Failure Assessment (SOFA) Score were measured in mice, and innate immune cell number/function in the blood and peritoneal cavity were assessed.

### Results

CLP significantly increased circulating levels of human resistin. The severity of sepsis-induced leukopenia was comparable between Retn+ and Rko mice. Resistin was associated with increased production of neutrophil reactive oxygen species, a decrease in

**Funding:** This work was supported by the National Institute of General Medical Sciences (NIGMS,

https://www.nigms.nih.gov) of the National Institutes of Health (NIH). Grant number K08 GM138825 (ASB). The funders had no role in study design, data collection and analysis, decision to publish, or preparation of the manuscript.

**Competing interests:** The authors have declared that no competing interests exist.

circulating neutrophils at 6 h and an increase in peritoneal Ly6C$^{hi}$ monocytes at 6 h and 24 h post-sepsis. However, intraperitoneal bacterial growth, organ dysfunction and mouse survival did not differ with resistin production in septic mice.

## Significance

*Ex vivo* resistin-induced impairment of neutrophil function do not appear to translate to increased sepsis severity or poorer outcomes *in vivo* following CLP.

## Introduction

Several recent studies have uncovered a strong correlation between blood concentrations of resistin and the severity of organ dysfunction in critically ill, septic patients [1–6]. While the physiologic role of human resistin remains unknown, it is thought to be a pro-inflammatory cytokine that triggers TNFα-independent intracellular signaling via the NFκB pathway [7]. Some investigators have demonstrated higher circulating resistin concentrations in sepsis non-survivors [3], whereas others have been unable to confirm this relationship [1, 5, 6, 8]. Previous studies from our laboratory reported that human resistin impairs neutrophil migration, decreases the production of reactive oxygen species required to kill intracellular and extracellular bacteria, and diminishes the production of neutrophil extracellular traps [9–11]. These resistin-mediated effects are concentration-dependent and reversible upon removal of resistin from the extracellular medium [9]. They also raise the possibility of using resistin not only as a diagnostic biomarker in sepsis but also as a therapeutic target in patients experiencing severe disease. However, the translational importance of our previous *in vitro* findings was limited by the lack of *in vivo* correlates.

Mice possess four resistin-like molecule (RELM) genes, while humans possess two of these genes, *Retn* and *Retnlb* [12–14]. An important difference between murine and human resistin is the completely different expression pattern, with the 12.5 kDa human resistin variant being expressed predominantly by macrophages [15, 16] instead of adipocytes. Furthermore, murine resistin is suppressed by TNFα and other proinflammatory cytokines, whereas human resistin is induced by TNFα [17]. Lazar *et al.* demonstrated that *BAC-Retn* mice generated on a murine *Retn*$^{-/-}$ background have circulating resistin levels that are comparable to those measured in humans under normal conditions, and that lipopolysaccharide (LPS) markedly increased resistin production by macrophages [15]. This model was used to demonstrate that resistin may actually exert a protective effect in LPS-induced shock [18]. Contrary to these findings, however, Jiang *et al.* reported resistin-induced pro-inflammatory neutrophil activation, neutrophil extracellular trap formation and increased severity of acute lung injury [19].

The significance of these opposing findings is limited as these studies have not yet been replicated in more clinically relevant models of sepsis. LPS injection has a greater LD$_{50}$ in mice than it does in humans. Furthermore, preparation purity of LPS often varies between studies and may cause different temporal cytokine responses as compared with surgical models of sepsis such as cecal ligation and puncture (CLP) [20–23]. We hypothesized that fecal peritonitis and subsequent surgical sepsis would trigger the production of human resistin in transgenic mice expressing this protein. We anticipated that sepsis-induced increases in human resistin would exacerbate the hypothermia, organ dysfunction and mortality in mice producing this cytokine, validating the translational relevance of our model. Confirmation of these findings might also provide an explanation for the clinically observed and concentration-dependent

relationship between resistin and organ dysfunction in sepsis. To enhance the translational relevance of our investigation, we employed the modified Mouse Severity of Sepsis (MSS) scoring system as a surrogate of the Sequential Organ Failure Assessment (SOFA) used in human patients [24, 25]. The SOFA score is used to classify sepsis severity based on indicators of organ dysfunction [26, 27]. The modified MSS scoring system has been shown to reliably predict the development of septic shock and death in a fecal peritonitis model of murine sepsis [24].

## Materials and methods

### Animal care

All experiments were approved by the Institutional Animal Care and Use Committee at the Pennsylvania State University College of Medicine (#201800418) and adhered to National Institutes of Health (NIH) guidelines. Information contained in the Materials and Methods section addresses the ARRIVE guidelines for the use of animals in research [28]. All surgeries were performed under general inhaled anesthesia with isoflurane, and all animals were premedicated with buprenorphine for postoperative analgesia. Male C57BL/6 mice, aged 8–12 weeks, were housed in the pathogen-free, animal care facility at the Pennsylvania State College of Medicine. They were maintained in a controlled environment (i.e., 23°C, 12 h/12 h light/dark cycle, and 30–70% humidity) in plastic cages with corncob bedding. Mice were provided water ad libitum and standard rodent chow (Teklad Global 2018, Envigo; Indianapolis, IN). Only male mice were used in the current study due to the well-described, gender-dimorphic response to CLP in mice [29].

### Mice

C57BL/6 mice transgenic for the human resistin gene ($Retn^{+/-/-}$, or Retn+) were obtained from Dr. Mitchell Lazar's lab via the University of California at Riverside. These mice were generated on a $Retn^{-/-}$ (resistin knockout, or Rko) background, as previously described [15]. The human resistin gene, along with 21,300 bp upstream and 4,248 bp downstream of its start site, was inserted through a bacterial artificial chromosome to generate a mouse line which was transgenic for the human resistin gene and whose macrophages express human resistin. Genome insertion of *hRETN* in the transgenic mouse lines was determined by Dartmouse, which sequences and analyzes thousands of SNPs throughout the mouse genome. All mice used had their genotype experimentally confirmed by using 1% agarose gel electrophoresis, following polymerase chain reaction amplification of the human resistin gene and of the housekeeping gene, GAPDH. Basal metabolic conditions of both mouse strains have been previously published [15].

### Sepsis model

Mice were randomly assigned to either septic or time-matched control groups. Polymicrobial peritonitis was induced using CLP, and separate groups of mice were studied either 6 h or 24 h following the procedure [30]. To minimize any possible circadian-induced changes, all experiments were initiated between 0700 and 1000 each morning. On Day 0, mice were anesthetized with isoflurane (3–5% induction with 2–3% maintenance; Vedco, St. Joseph, MO) with oxygen. Using sterile surgical technique, a midline laparotomy incision was made after shaving and cleaning the abdomen with betadine solution. The cecum was ligated using surgical silk (4–0, Covidien, Minneapolis, MN) approximately 1 cm from the distal end, and punctured twice using a 25-gauge needle. Patency was ensured by extruding a small amount of feces from

the puncture sites before replacing the cecum back into the abdominal cavity. The abdominal wall was closed using 4–0 silk suture prior to closing the skin incision with wound clips. Resuscitation consisted of 1 mL of warmed, sterile, normal saline containing 0.05 mg/kg buprenorphine (Reckitt Benckiser Pharmaceuticals, Richmond, VA) and was administered subcutaneously 15 min prior to surgery. Body temperature was maintained constant during surgical procedures by keeping mice on a warming pad until they regained consciousness. Sham-control mice underwent the same surgical procedure except the cecum was not ligated or punctured. Mice were euthanized at 6 or 24 ± 2 h after laparotomy ± CLP and this is referred to as the "acute sepsis" group.

A separate group of mice was used to investigate alterations in the recovery phase post-sepsis. Starting 6 h post-CLP (Day 1), antibiotic (0.5 mg meropenem; Fresenius Kabi, Lake Zurich, IL) and buprenorphine were injected subcutaneously (total volume 1 mL of sterile 0.9% sterile saline) every 8–12 h for the next six days. Time-matched control mice received the same volume of fluid resuscitation with buprenorphine and antibiotic. Death was not considered an endpoint in these survival analyses. Rather, pre-determined humane endpoints and Body Condition Scoring were used to determine the appropriateness of euthanasia [31]. Mice were checked twice per day and euthanized if they were found to be moribund, immobile, unable to consume food and/or water, had >20 percent weight loss (as compared to weight at the beginning of the experiment), prolonged diarrhea, significantly labored breathing, post-surgical wound dehiscence or visible infection at the surgical site, tremors or paralysis, a rectal temperature < 27°C for 12 h or more, or a Body Condition Scoring of ≥2.

## Sampling

Serial blood samples were collected via cheek bleeding targeting the submandibular and retro-orbital veins. Whole blood was collected at 0 h, 6 h and 24 h following surgery in micro-capillary EDTA blood collection tubes (RAM scientific, Nashville, TN), for complete and differential blood counts analysis on Heska Element HT5 Veterinary Hematology Analyzer (Heska Corp, Loveland, CO). Manual leukocyte cell differential counts were computed by dedicated veterinary lab technicians following Diff-Quick staining (Baxter, Detroid, MI). In a separate cohort of animals, whole blood was collected at the predetermined time points, allowed to clot at room temperature for 30 min, and subsequently centrifuged to isolate the serum fraction for analysis of blood glucose, creatinine, blood urea nitrogen (BUN), aspartate aminotransferase (AST), alanine aminotransferase (ALT) and total bilirubin concentrations. Sample analysis was performed on COBAS MIRA Plus Chemistry System (Roche Diagnostic Systems Inc., Somerville, NK).

Peritoneal fluid was collected either 6 h or 24 h post-surgery, as previously described but with minor modifications [32]. Briefly, following general anesthesia and cervical dislocation, the abdominal incision was reopened under sterile conditions. The first lavage was performed with a 1mL of warmed Hank's Balanced Salt Solution (HBSS, Gibco, Grand Island, NY), and used for microbial culture. This lavage was followed by a further 24mL of the same solution, in divided doses. 100μL of the 1mL lavage was used for bacterial cultures. Cells from the remaining 900μL of peritoneal fluid were pooled with cells obtained from the subsequent 24mL lavage and used for flow cytometry.

## Flow cytometric analysis

Samples of 100μL of whole blood as well as peritoneal cells obtained as described above were separately stained with antibodies. Cell suspensions were blocked with DPBS/2% FBS containing 0.5mg/mL Fc Block (BD Bioscience) for 10 min at 4°C. Blocked cells were stained with

fluorochrome conjugated monoclonal antibodies obtained from BD Bioscience–CD11b (M1/70), CD45.2 (104), F4/80 (T45-2342), Ly-6G (1A8), Biolegend–CD90.2 (30-H12) or Thermo Fisher Scientific–MHC class II (I-A/I-E). Cells were washed in DPBS alone and then incubated with a Fixable Viability Dye eFluor® 780 (eBioscience) for 20 min at 4˚C to discriminate viable from non-viable cells. Red blood cells were lysed with a red blood cell lysis buffer (Tonbo Biosciences) either before (peritoneal cells) or after (whole blood) antibody staining. Cells were then washed with DPBS with 2% FBS and 0.09% sodium azide (Becton, Dickinson and Company, Sparks, MD). Quantification of cells from whole blood and peritoneal washings was achieved by using Countbright Plus Absolute Counting Beads (Thermo-Fisher Scientific, Waltham, MA). Data were collected using a BD LSRII flow cytometer and analyzed using FlowJo version 10.7.1 (Becton, Dickinson and Company, Franklin Lakes, NJ) using the gating strategy shown in (S1 Fig). Gating of monocytes for F4/80 positive cells, in order to identify the macrophage subset, was performed only in peritoneal cells and not in blood cells.

## Physiologic measurements and assessment of sepsis severity

Rectal temperature was measured with a digital probe (Harvard Apparatus; Holliston, MA). Heart rate, peripheral oxygen saturation and breath rate were measured by the non-invasive, MouseOx small animal pulse oximeter neck collar (Starr Life Sciences Corp; Oakmont, PA). Breath rate was additionally confirmed by manual counting. To ensure consistency in the depth of anesthesia used for monitoring vital signs, anesthesia was maintained by using 2–3% inhaled isoflurane, following at least 2 min of anesthetic induction with 4–5% isoflurane and a negative response to toe pinch.

## Cytokine measurements

Serum concentrations of human resistin at 6 h and 24 h were measured by human resistin quantikine ELISA assay at 1:5 serum dilution (R&D systems, Minneapolis, MN). This assay is specific for the detection of human resistin, with < 50% cross-species reactivity reported by the manufacturer. Additionally, serum concentrations of angiopoietin-2, CXCL2, IL-1β, IL-10, IL-27, SP-D, TNFα, CXCL1, ICAM-1, IL-6, IL-12 p70, RAGE, P-selectin, syndecan-1 and uPAR were measured on blood samples collected at 6 h and 24 h by a magnetic bead-based multiplex assay (R&D systems, Minneapolis, MN) after a 1:2 dilution.

## Quantification of bacterial growth in peritoneal samples

To compare bacterial killing capacity between Rko and Retn+ mice, peritoneal fluid samples taken from mice at 6 h and 24 h post-sepsis were subjected to aerobic, microbial cell culture by using methods previously described [32]. Briefly, 100μL of peritoneal fluid obtained after lavage of the peritoneal cavity with 1mL of warmed HBSS underwent serial dilution followed by plating on 5% sheep's blood agar plates. The number of colony-forming units (CFUs) at 18 h following bacterial plating was calculated based on colony counts and dilution factor.

## Neutrophil isolation and assessment of reactive oxygen species production

Preliminary experiments identified low neutrophil counts and poor neutrophil viability of neutrophils isolated from peritoneal washings following CLP. Therefore, to assess neutrophil function, we utilized density gradient separation of cells obtained from the bone marrow. Mice were sacrificed at 24 h following CLP and their bone marrow was carefully layered onto a density gradient created using room temperature Histopaque 1077 and 1119 (Sigma-Aldrich Inc, St. Louis, MO) [33]. The neutrophil fraction was washed twice in PBS, and $1 \times 10^5$ viable

cells were incubated for 2 h at 37˚C and 5% $CO_2$ in complete RPMI with 100 ng/mL of LPS from *E. coli* O55:B5 or 100ng/mL of Phorbol Myristate Acetate (PMA), both from Sigma-Aldrich Inc. Control neutrophils were incubated with an equal volume of PBS. 7.5μM of Cell-ROX Deep Red reagent (Thermo-Fisher Scientific) in complete RPMI medium was then added to each vial and further incubated at 37˚C for 30min in the dark. Following cell washing in PBS, cells were fixed in 4% paraformaldehyde and fluorescence was assessed using a Flexstation 3 multiplate reader (Molecular Devices LLC, San Jose, CA) using Softmax Pro 7 (Molecular Devices LLC). Change in fluorescence intensity between stimulated and control cells was calculated as a comparable measure of reactive oxygen species production by neutrophils.

## Statistical analysis

All data were analyzed using GraphPad Prism 9.1.2 (San Diego, CA) by two-way or three-way analysis of variance (ANOVA, genotype x sepsis x time) or mixed effects analysis matched on genotype. Tukey or Sidak corrections for multiple comparisons were used when appropriate, and values considered statistically significant when P< 0.05. Unpaired t tests and nonparametric tests with two-tailed alpha value were used to compare Rko and Retn+ results at the same time interval. Where appropriate, figures represent mean with standard error, or have individual data points presented as a box-and-whisker plot with the mean, upper and lower quartiles, and upper and lower values identified. Sample sizes are detailed in figure legends.

## Results

### General characteristics of septic mice

In total, 107 experimental mice were studied, of which 59 were Retn+ and 48 were Rko. Mean weight, at baseline, was 23.9 ± 2.2 g for Retn+ mice and 23.6 ± 2.0 g for Rko mice. In Retn + mice, sepsis significantly increased blood resistin concentration as compared with control mice undergoing sham surgery only (P = 0.01 at 6 h and P = 0.003 at 24 h). Rko mice did not demonstrate an increase in blood resistin concentration following the onset of sepsis (P = 0.62 at 6 h and P = 1.0 at 24 h). There was no significant difference in the blood resistin concentration in Retn+ mice between 6 h and 24 h after CLP (P = 0.36) (Fig 1). The average decrease in

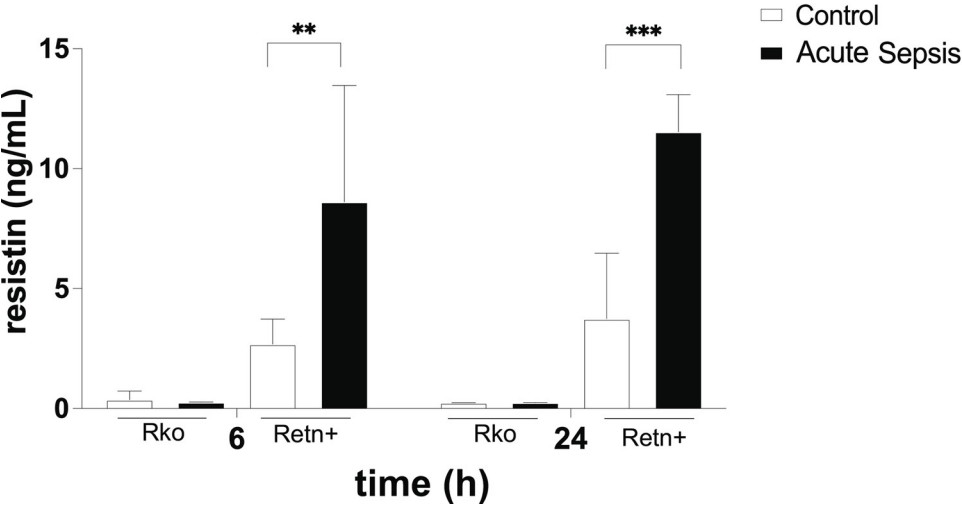

**Fig 1. Serum concentrations of human resistin in hRetn-/- mice (Rko) and in hRetn-/- mice transgenic for the human resistin gene (Retn+).** Concentrations were measured at 6 h and 24 h post-surgery. n = 3 for Rko with acute sepsis at 6 h and Retn+ with acute sepsis at 24 h; n = 4 for Rko acute sepsis and control at 24 h; n = 6 for Retn+ acute sepsis and control at 6 h; n = 6 for Retn+ control at 24 h; n = 7 for Rko control at 6 h.

body weight within the first 24 h following surgery was 1.63 ± 0.30 g in Retn+ mice with acute sepsis and 2.06 ± 0.35 g in Rko mice with acute sepsis (P = 0.36).

As most previous reports showing a positive relationship between the blood resistin concentration and sepsis severity have utilized a critical illness scoring system such as the Acute Physiology and Chronic Health Evaluation II [34] or SOFA score (on which a diagnosis of sepsis is now based [27]), we chose to focus on the effect of resistin on equivalent components of the SOFA score in mice. The SOFA score is based on the adequacy of blood oxygenation (a surrogate of respiratory dysfunction), platelet count (hematologic dysfunction), total blood bilirubin concentration (hepatobiliary dysfunction), blood pressure (cardiovascular dysfunction), degree of encephalopathy (cerebral dysfunction) and blood creatinine concentration (renal dysfunction). Equivalent values that were measured in mice included platelet count, bilirubin concentration, ALT, AST and blood creatinine concentration, heart rate as a surrogate of cardiovascular function, and breath rate and $SpO_2$ as surrogates of respiratory function.

Serum biochemistry analysis revealed no significant difference in BUN, creatinine, bilirubin, AST or ALT in the 24 h samples from Retn+ or Rko mice (S2 Fig). However, acute sepsis did decrease the serum glucose concentration in both groups of mice, although there was no difference in the severity of hypoglycemia between Retn+ and Rko mice (S2 Fig) [15]. Mixed-effect analysis of leukocyte count from the analysis of blood peripheral smears revealed a time-dependent effect (P<0.0001) and a sepsis-dependent effect (P<0.0001) causing progressive leukopenia (Fig 2). However, leukocyte count was independent of resistin concentrations at 6 h and 24 h following surgery (P = 0.37).

## Resistin is associated with a decrease in circulating neutrophils, and with an increase in peritoneal Ly6C$^{hi}$ monocytes at 6 h post-sepsis

Given the paucity of white blood cells in acute sepsis, we performed flow cytometric analysis of blood and peritoneal fluid samples at 6 h and 24 h to quantify differences in leukocyte cell

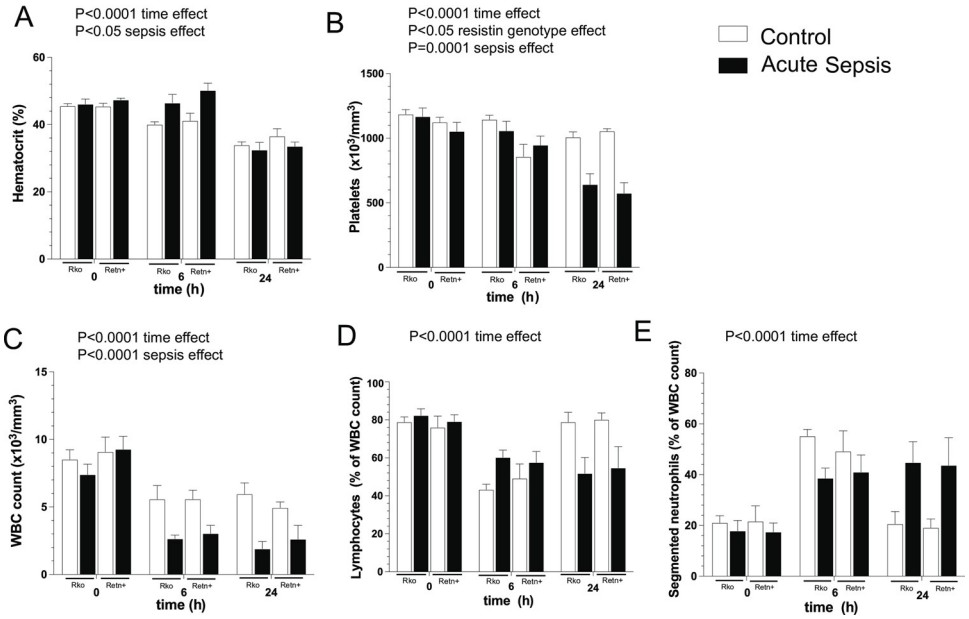

**Fig 2. Complete blood count and leukocyte differential analysis at 6 h and 24 h post-surgery, in resistin knockout C57BL/6 mice (Rko) and transgenic C57BL/6 mice on an Rko background (Retn+).** A—E represent mean (+/-SEM) serum concentrations of the named parameters following surgery. For A-C, n = 6 per time point per group. For D and E, n = 5–7 and results reflect cell staining followed by manual cell differential count. P values represent statistically significant results of ordinary 3-way ANOVA (time x resistin genotype x sepsis) within each cell type.

populations more accurately. At 6 h following sepsis, the number of circulating neutrophils
(P = 0.0063) and antigen-presenting neutrophils (P = 0.0052) was significantly lower in mice
producing resistin (Fig 3A). This decrease in blood neutrophil count did not appear to corre-
late with an increase in peritoneal neutrophil count (Fig 3C). Sepsis was also associated with
neutropenia at 24 h following surgery, although resistin production did not significantly affect
the degree of neutropenia at this time point (Fig 3E). In contrast to 6 h (and as expected in the
setting of acute infection) peritoneal neutrophil count increased at 24 h indicating transmigra-
tion to the site of infection. Resistin did not, however, cause a significant difference in the
degree of transmigration into the peritoneal fluid (Fig 3G). Neither did we detect a difference
in the proportion of antigen-presenting neutrophils in the blood or peritoneal fluid of at 24 h
in resistin-producing mice (Fig 3E).

Consistent with changes expected during acute sepsis, the peritoneal fluid of septic mice
had a an increased Ly6C$^{hi}$ monocyte count (Fig 3D). Ly6C$^{hi}$ monocytes are a heterogeneous
population of leukocytes (corresponding to 'classic' monocytes in humans) which are rapidly
recruited to sites of inflammation where they mediate phagocytosis. Interestingly, resistin pro-
duction was associated with higher Ly6C$^{hi}$ monocyte counts both at 6 h following sham sur-
gery (P = 0.026) and at 6 h following sepsis (P = 0.0062). Increased numbers of peritoneal
inflammatory Ly6C$^{hi}$ monocytes persisted at 24 h following sepsis (P = 0.034).

## Resistin production does not affect serum concentrations of ancillary inflammatory cytokines at 6 h and 24 h following onset of acute sepsis

Three-way ANOVA revealed time-dependent decreases in the serum concentrations of
CXCL-1 (P = 0.018), CXCL-2 (P = 0.041), IL-1β (P = 0.035), and IL-6 (P = 0.0086), as well as a

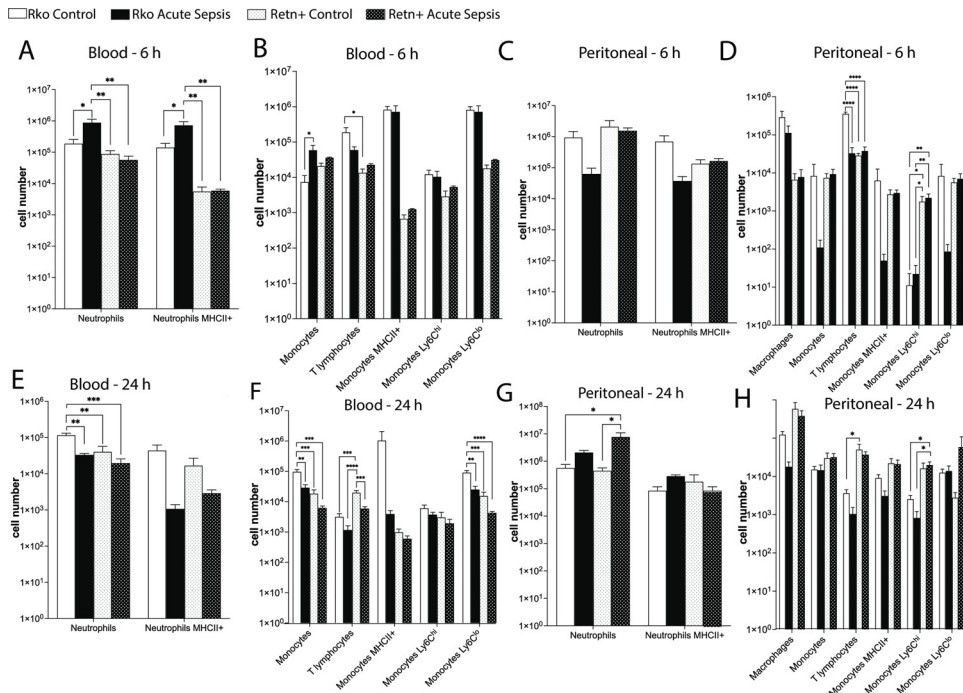

**Fig 3. Flow cytometry assessment of cellular differential at 6 h and 24 h post-cecal ligation and puncture, in
knockout C57BL/6 mice (Rko) and mice expressing human resistin (Retn+).** A, Blood neutrophils at 6 h; B, Blood
monocytes and T lymphocytes at 6 h; C, Peritoneal neutrophils at 6 h; D, Peritoneal monocytes, macrophages and T
lymphocytes at 6 h; E, Blood neutrophils at 24 h; F, Blood monocytes and T lymphocytes at 24 h; G, Peritoneal
neutrophils at 24 h; H, Peritoneal monocytes, macrophages and T lymphocytes at 24 h. n = 4 for Rko at 6 h and 24 h,
and for Retn+ at 24 h. n = 3 for Retn+ at 6 h.

time-dependent increase in angiopoietin-2 (P<0.0001) following surgery (Fig 4). Sepsis increased the serum concentrations of angiopoietin-2 (P<0.0001) and ICAM-1 (P = 0.0043), although the magnitude of increase did not differ statistically between Rko and Retn+ mice. Resistin did not alter the serum concentrations of SP-D, angiopoietin-2, CXCL2, IL-1β, IL-10, IL-27, TNFα, CXCL1, ICAM-1, IL-6, IL-12 p70, RAGE, P-selectin, syndecan-1 or uPAR at 6 h or 24 h following the onset of acute sepsis (S1 and S2 Tables in S1 File).

## Resistin production is not associated with decreased survival in acute sepsis

A survival analysis was performed comparing a cohort of mice undergoing CLP with time-matched controls. MSS and physiologic parameters were assessed twice daily, at which time mice were given warmed normal saline subcutaneously, as well as antibiotics (0.5 mg) and analgesics (0.05mg/kg buprenorphine). Mice meeting humane endpoints for euthanasia were considered dead for purposes of assessing survival. The survival study was terminated 7 days post-surgery, with any surviving mice being euthanized at that time point. Median survival for Retn+ mice with acute sepsis was 82 h, as compared with 78 h for Rko mice with acute sepsis (Fig 5A). Survival results were not significantly different when compared by Log-rank (Mantel-Cox) test (P = 0.408) and Gehan-Breslow-Wilcoxon test (P = 0.726).

## Resistin production does not affect bacterial growth from peritoneal fluid during acute sepsis, but increases reactive oxygen species production by neutrophils

Resistin blocks the bactericidal activity of cultured neutrophils through partial reduction of F-actin polymerization and suppression of the oxidative burst [10]. Furthermore, removal of

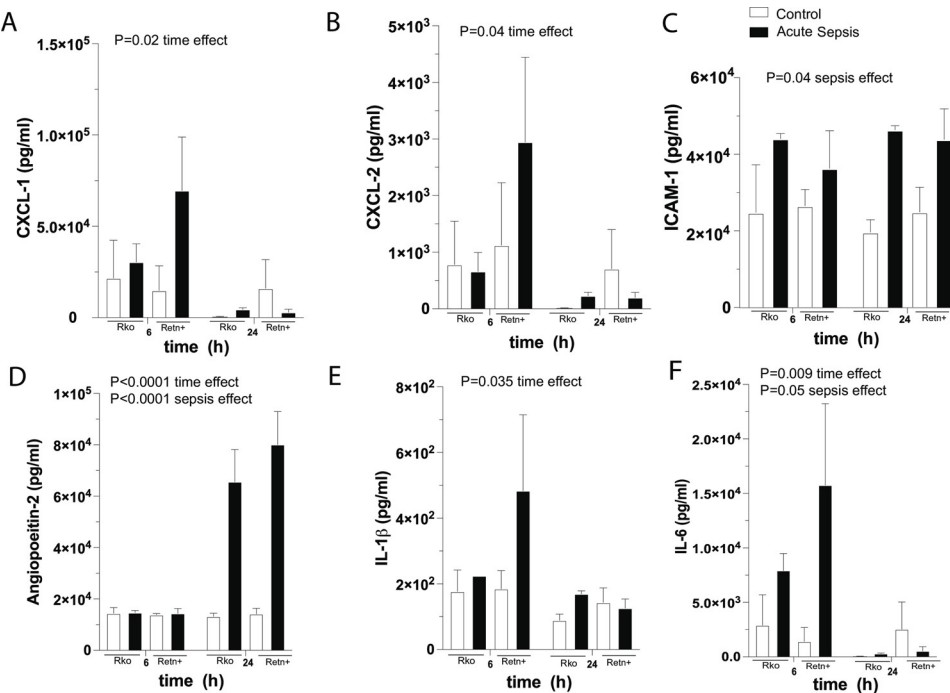

**Fig 4. Serum concentrations of human cytokines in hRetn-/- mice (Rko) and mice expressing human resistin (Retn+).** Concentrations were measured at 6 h and 24 h post-CLP. n = 4 for Retn+ at 6 h and 24 h, n = 2 for Rko at 6 h and n = 4 for Rko at 24 h. P values for statistically significant relationships on ordinary 3-way ANOVA (time versus sepsis effect versus genotype) are shown.

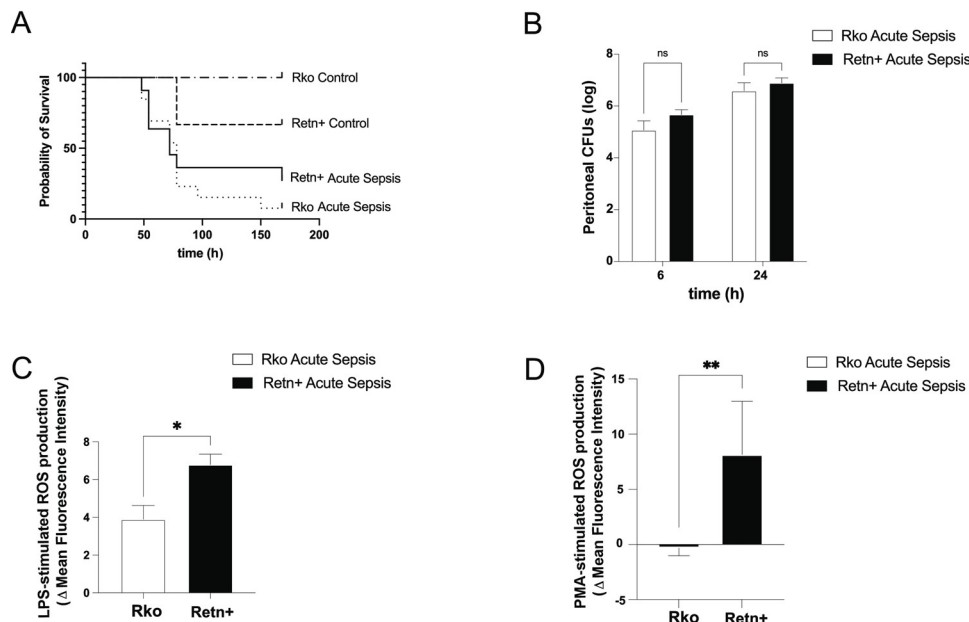

**Fig 5. Outcomes for knockout mice (Rko) and those producing human resistin (Retn+) in acute sepsis induced by CLP.** A, Kaplan-Meier curve following acute sepsis (time = 0 h), with n = 14 for Retn+ of whom 11 underwent CLP and 3 underwent sham surgery; n = 18 for Rko of whom 13 underwent CLP and 5 underwent sham surgery. High apparent mortality in sham-operated Retn+ is attributed to one mortality in a group with n = 3, which we surmise to be an outlier. B, Peritoneal bacterial load in relation to expression of human resistin. There was no bacterial overgrowth in Rko as compared with Retn+ mice with sepsis. n = 4 for Rko and n = 3 for Retn at 6 h; n = 9 for Rko and Retn+ at 24 h. C, Change in fluorescence intensity between neutrophils stimulated with PMA and unstimulated cells following exposure to CellROX reagent at 24 h following CLP (n = 6 for Rko and Retn+ mice). D. Change in fluorescence intensity between neutrophils stimulated with LPS and unstimulated cells following exposure to CellROX reagent at 24 h following CLP (n = 6 for Rko and Retn+ mice).

inhibitory resistin from extracellular media reverses neutrophil dysfunction, restores neutrophil migration and their ability to generate reactive oxygen species, and reconstitutes intracellular bacterial clearance [9]. In contrast, resistin does not appear to affect the ability of macrophages or monocytes to kill either Gram-positive or Gram-negative organisms [10]. Given the decreases in blood and peritoneal neutrophil concentrations observed at 6 h in resistin-producing septic mice, we investigated bacterial growth and neutrophil reactive oxygen species production at this time point and at 24 h following the onset of sepsis. The peritoneal cavity of mice sacrificed at 6 h or 24 h was lavaged with warmed normal saline, and serial dilutions of peritoneal fluid were used to establish countable colonies of aerobic bacterial cultures. These were measured 18 h following dilution and plating onto 5% sheep's blood agar plates. The peritoneal cavity of mice undergoing sham surgery was sterile. There was no relationship between resistin production and the number of CFUs at 6 h and 24 h post-CLP (Fig 5B). We further assessed the ability of neutrophils to generate reactive oxygen species in mice producing resistin, to replicate our prior *in* vitro findings. We did not observe a significant difference between fluorescence emission at 30 and 60 min following CellROX stimulation (S3 Fig). Congruent with prior *in vitro* results, neutrophils isolated from resistin-producing mice generated higher concentrations of reactive oxygen species than those isolated from resistin-deficient mice, both following stimulation with LPS (P = 0.0016, Fig 5C) and PMA (P = 0.01, Fig 5D).

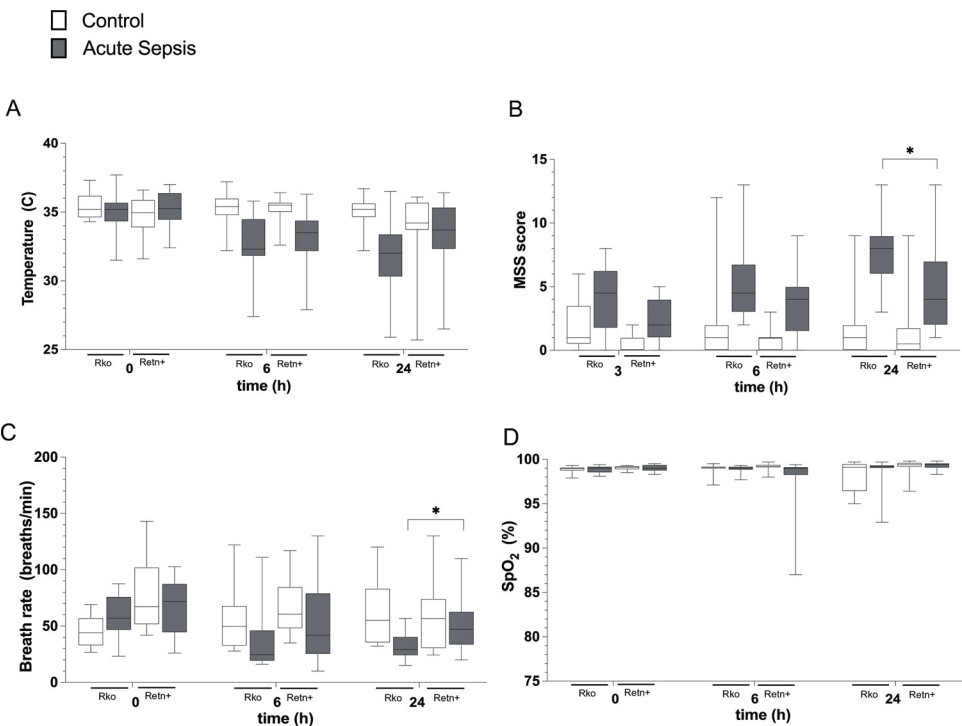

**Fig 6. Physiologic parameters at 0 h, 6 h and 24 h post-surgery in mice lacking resistin (Rko) and in mice producing human resistin (Retn+).** A, Rectal temperature, B, Mouse Severity of Sepsis Score; C, Breath Rate; D, Oxygen saturation by pulse oximetry. N = 10–25 per group. Box and violin plot line at mean value for each group.

### Resistin production is associated with higher MSS score and tachypnea at 24 h following bacterial peritonitis, but no difference in body temperature

In addition to the SOFA-derived parameters described above, we measured temperature and modified MSS scores as independent surrogates of illness severity in murine sepsis. Mai *et al* have compared the MSS, M-CASS and MGS scoring systems within 24 h of CLP sepsis and have reported that modified MSS score combined with body temperature are effective markers of disease severity and reliably predict death following sepsis secondary to CLP [25].

Septic mice developed significant and progressive hypothermia as compared with control mice (P<0.0001) (Fig 6A). A time-dependent decrease in heart rate and breath rate was seen in septic mice (Fig 6C), although peripheral blood oxygenation (SpO$_2$) remained unchanged (Fig 6D). As expected, mice with acute sepsis had a higher mean modified MSS score at 6 h and 24 h as compared with control mice undergoing sham surgery (P<0.0001) (Fig 6B). At 24 h, resistin-producing mice had elevated MSS scores (P = 0.01) more tachypnea (P = 0.048), although their temperature was not significantly different from that of resistin-deficient mice (Fig 6).

### Discussion

The primary goal of this study was to establish a reproducible *in vivo* model with which to investigate the effect of human resistin on sepsis severity and organ dysfunction. The clinical relevance of this objective lies in the continued lack of clarity regarding the pathophysiologic role of resistin in human sepsis, an issue which has been debated since 2007 [1]. Since this initial report, certain studies have confirmed the value of resistin as a biomarker of sepsis severity

[4, 5, 35–40] while others have suggested a more direct relationship between blood resistin concentration, endothelial markers and organ failure [8]. Our previous studies, however, showed that physiologically relevant concentrations of resistin directly inhibit neutrophil function *ex vivo*, indicating suppression of innate immunity as a likely pathophysiologic mechanism of resistin [9, 11]. We demonstrated that resistin inhibits bacterial killing (by cell-cultured neutrophils) of the Gram-negative bacterium *Pseudomonas aeruginosa* and of the Gram-positive bacterium *Staphylococcus aureus* which are common pathogens causing healthcare-acquired infections and sepsis [10].

These encouraging data provided a likely explanation by which elevated resistin concentrations could mediate severe sepsis and shock *in vivo*. In the current study we selected an established transgenic mouse model in which circulating human resistin concentrations were within the normal human range and markedly increased upon exposure to endotoxin due to production by mouse macrophages [15]. This same model has been used to demonstrate that resistin plays a protective role in LPS-induced septic shock, with increased survival of helminth-infected, resistin-producing mice after a rapidly lethal dose of LPS [18]. However, as the relevance of acute endotoxemia as a model of clinical sepsis has been questioned [41], our study is the first to report the effect of resistin in a well-established and clinically relevant surgical model of murine abdominal sepsis. Mice were provided with analgesics, antibiotics and fluid therapy analogous to the care patients receive during acute abdominal sepsis. Furthermore, our study is the first (to our knowledge) to apply practical, SOFA-derived parameters in mice alongside the recently validated MSS score, to enhance the translational relevance of our results.

We did not detect a significant difference in SOFA-based laboratory markers of organ dysfunction, including bilirubin, creatinine concentration or platelet count within 24 h of sepsis onset in Retn+ mice. We did, however, note that an increase in modified MSS score and a decrease in core body temperature following acute sepsis independently portended poorer outcomes, confirming previously reported findings [25]. Thus, our data support the continued use of these endpoints as factors guiding humane endpoints for rodent euthanasia in acute sepsis. The current study also provides further evidence that resistin is more likely to be a marker of acute inflammation rather than a disease-mediating cytokine in acute sepsis. Sepsis predictably increased serum concentration of acute, proinflammatory cytokines, such as IL-6, as well as markers of endothelial dysfunction markers (ICAM-1 and angiopoietin 2). However, there was no relationship between blood resistin concentration and that of ancillary, sepsis-induced cytokines. Sepsis is known to cause widespread endothelial dysfunction, and angiopoietin-2 has recently gained prominence as an alternative biomarker of sepsis severity. By destabilizing the vascular endothelium, angiopoietin-2 causes vascular leak and interferes with the microvascular control of blood flow [42–45]. The dramatic increase in angiopoietin we observed in mice undergoing CLP suggests that the murine CLP model may be a relevant model with which to further investigate the role of angiopoietin-2 in humans.

Our results indicate that resistin may increase neutrophil transmigration into the peritoneal cavity and cause more robust neutrophil oxidative burst. Resistin production was found to correlate with an increase in peritoneal fluid Ly6C$^{hi}$ (CX3CR1$^{int}$CCR2$^{+}$) monocytes at 6 h post-sepsis and a decrease in blood Ly6C$^{lo}$(CX3CR1$^{hi}$CCR2$^{−}$) monocytes at 24 h post-sepsis. While both monocyte subsets are involved in pro- and anti-inflammatory responses, the latter monocyte subset have been dubbed 'patrolling' monocytes and they correspond to human 'non-classic' monocytes. Adhering to the luminal side of the endothelium, they scavenge luminal debris and maintain endothelial integrity although their roles are still poorly understood [46]. Ly6C$^{hi}$ monocytes form an integral part of sterile and infectious inflammation, and our data indicate their increased presence in the peritoneal cavity of resistin-producing mice. The significance

of this finding is uncertain in the context of similar macrophage numbers and peritoneal bacterial growth in resistin-producing and resistin-deficient mice. Murine monocytes possess tremendous plasticity, however, and blood Ly6C$^{hi}$ monocytes can differentiate into Ly6C$^{lo}$ monocytes or into monocyte-derived dendritic cells or macrophages in infected tissue (in cases where tissue macrophages are depleted and need to be regenerated) [46, 47]. This activation and differentiation may not yet have occurred at our observed time points. Alternatively, while the numbers of Ly6C$^{hi}$ monocytes are increased in resistin-producing mice, the cells may be dysfunctional and unable to execute their proinflammatory and/or phagocytic functions. The unchanged proportion of antigen-presenting monocytes with resistin production would corroborate this hypothesis. Further studies are required into the role of human resistin on septic Ly6C$^{hi}$ monocytes. Given that human resistin is produced by monocytes in Retn + mice, one may also postulate whether inflammation-triggered resistin release is causing an autocrine, detrimental effect on peritoneal Ly6 h$^{i}$ monocytes transmigrating into the peritoneal cavity.

The findings of our current investigation, and their apparent discordance with prior *in vitro* findings, underline the importance of using a clinically relevant model when performing translational investigations of human disease to delineate the role of biologic mediators on complex temporal immune responses in sepsis. Despite differences in blood and peritoneal fluid cell profiles, we did not observe a difference in our primary outcomes between mouse cohorts. Our results raise the possibility of experimental model artifact, as they contrast with prior (albeit conflicting) data regarding protective [18] and detrimental [19] effects of resistin following LPS exposure. It is also possible that our experimental model provides further evidence of the caution required in extrapolating results induced from murine endotoxemia research to human sepsis [41]. The inability of the scientific community to decipher a consistent role of human resistin in sepsis since its discovery in 2001, beyond that of an inflammatory biomarker like many others, may itself question the clinical relevance of resistin in this disease.

## Supporting information

**S1 Fig. Flow cytometry gating strategy.** Representative flow cytometry profiling for peritoneal fluid analysis obtained 24 h after acute sepsis.
(TIF)

**S2 Fig. Organ injury biomarkers, in resistin knockout (Rko) and transgenic C57BL/6 mice on an Rko background (Retn+).** A—F represent mean (+/- SEM) serum concentrations of the named biomarkers at 24 h following surgery, of which only glucose concentration varies among control and acute sepsis (p = 0.009 for Retn+ mice; p = 0.005 for Rko mice). n = 9–13 per group.
(TIF)

**S3 Fig.** Kinetics of CellROX Deep Red Reagent, from 5 min to 65 min following the addition of CellROX reagent to neutrophils isolated from bone marrow of Rko (A, B) or Retn+ (C, D) mice. Values represent average fluorescence emission (minus control fluorescence) from 1.5 million cells primed with 100ng/ml PMA or LPS for 2 h at 38˚C.
(TIF)

**S1 File. Contains supporting S1 and S2 Tables, which detail the mean blood cytokine concentrations at 6h and 24h respectively following surgery.**
(DOCX)

## Acknowledgments

We thank Abigail Samuelsen for her excellent technical assistance, Dr. Daniel McKeone for his expertise with Luminex assays and Dr. E Scott Halstead for his expertise with flow cytometry. We thank Dr. Meera Nair from the University of California (Riverside, CA), for providing the Rko and Retn+ mice used in these experiments. We thank the members of PSU-College of Medicine Flow Cytometry Core Hershey (RRID:SCR_021134), for their assistance with sample processing.

## Author Contributions

**Conceptualization:** Anthony S. Bonavia, Zissis C. Chroneos, Victor Ruiz-Velasco, Charles H. Lang.

**Data curation:** Anthony S. Bonavia.

**Formal analysis:** Anthony S. Bonavia.

**Funding acquisition:** Anthony S. Bonavia.

**Investigation:** Anthony S. Bonavia.

**Methodology:** Anthony S. Bonavia, Zissis C. Chroneos, Victor Ruiz-Velasco, Charles H. Lang.

**Resources:** Zissis C. Chroneos, Victor Ruiz-Velasco, Charles H. Lang.

**Supervision:** Charles H. Lang.

**Validation:** Anthony S. Bonavia.

**Visualization:** Charles H. Lang.

**Writing – original draft:** Anthony S. Bonavia.

**Writing – review & editing:** Anthony S. Bonavia, Zissis C. Chroneos, Victor Ruiz-Velasco, Charles H. Lang.

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
