## [Decision Letter · Decision Letter 0]

16 Nov 2021

PONE-D-21-27883Organ dysfunction and outcomes in a mouse model of acute surgical sepsis are independent of resistin concentrationPLOS ONE

Dear Dr. Bonavia,

Thank you for submitting your manuscript to PLOS ONE. After careful consideration, we feel that it has merit but does not fully meet PLOS ONE’s publication criteria as it currently stands. Therefore, we invite you to submit a revised version of the manuscript that addresses the points raised during the review process.

Your manuscript was reviewed by two experts and we receive positive feedback with major comments. All comments must be addressed during revision.

We look forward to receiving your revised manuscript.

Kind regards,

Partha Mukhopadhyay, Ph.D.

Academic Editor

PLOS ONE

Journal Requirements:

2. Please amend your Methods section to state the method of euthanasia used in the mice studies

Reviewers' comments:

Reviewer's Responses to Questions

**Comments to the Author**

1. Is the manuscript technically sound, and do the data support the conclusions?

Reviewer #1: Partly

Reviewer #2: Yes

2. Has the statistical analysis been performed appropriately and rigorously? 

Reviewer #1: Yes

Reviewer #2: Yes

3. Have the authors made all data underlying the findings in their manuscript fully available?

Reviewer #1: Yes

Reviewer #2: Yes

4. Is the manuscript presented in an intelligible fashion and written in standard English?

Reviewer #1: Yes

Reviewer #2: Yes

5. Review Comments to the Author

Reviewer #1: In the present study “Organ dysfunction and outcomes in a mouse model of acute surgical sepsis are independent of resistin concentration”, the authors compared the resistin deficient mice and resistin transgenic mice in CLP model. Here are some major concerns to be addressed:

1. To confirm the function of human resistin in mice, the authors need to detect a classical signal of resistin in the transgenic mice in comparison to the background knockout mice.

2. Judging from the survival experiment, almost 25% of the sham operation transgenic mice reached euthanasia point. The percentage should be very low. The surgery itself may have introduced too much incomparable variation and the current research is not valid.

3. The resistin transgenic mice have better acute neutrophil response, better subacute macrophage response and better neutrophil function. 24 hours after sepsis induction, these mice stopped inflammatory reaction. The resistin deficient mice showed more disorganized immune response. However, the transgenic mice do not show better bacterial clearance or survival. The authors need to explain why.

4. The authors need to explain why resistin transgenic mice have low MHC-II expression.

5. The authors need to show basic metabolic conditions of the mice. Do resistin transgenic mice have lower body weight, less food intake, lower metabolism rate, or more bacterial growth in the cecum compared to resistin deficient mice?

6. The authors need to show sham-operational groups in figure 3.

7. Macrophages should not be in circulation. The authors need to correct interpretation of flow cytometry result.

8. The current study did not find any direct evidence of organ dysfunction. Also, this study compared high concentration of resistin versus almost no resistin. This is not a comparison between different concentration of resistin. The title should not include “organ dysfunction” or “concentration”.

Reviewer #2: In this manuscript Bonavia and his coworkers explored the in vivo physiological significance of the human resistin on sepsis severity and organ dysfunction using the cecal ligation and puncture surgical sepsis model on resistin knockout (Retn-/-) and transgenic mice expressing the human resistin gene and its regulatory elements on Retn-/- background. Since in the literature controversial data were published whether resistin is just a biomarker of sepsis severity and organ failure or it could be a therapeutic target, this manuscript has important findings that aims to clarify the in vivo clinical significance of resistin.

As for the formal aspects, the manuscript is logical, well-written and readable, however, there are some typos that should be corrected in the final paper: for instance, in the first sentence of the Results section mice and rats were both mentioned as experimental subjects, even though only mice were used in the present manuscript.

Although the goals are clear and the conclusions are in line with the presented Figures, there are some issues that needs to be addressed:

1. Authors should elaborate in the Introduction for a sentence why was it necessary to express the human resistin gene in resistin KO mice? Why they couldn`t simple use resistin WT and resistin KO mice for this study since they mention that the sequence homology between the human and murine gene is ~ 60% and similar transcriptional regulation is presumed. According to NCBI Gene the human and murine Retn have completely different expression pattern.

2. Since the human variant of Retn expressed primarily in macrophages, was this well-established mouse model Retn-/-/+ verified previously that it has similar expression pattern?

3. What can be known about the other 3 mouse resistin-like molecules? Can any of those contribute to the effects shown in these experiments? If not, why?

4. The authors detected significant differences in LPS and PMA stimulated peritoneal neutrophil ROS production measured by CellROX Deep Red reagent. Neutrophils were stimulated for 2 hours than incubated with CellROX for 30 min and finally a cumulative value was measured. Authors should measure and show kinetics of oxidative burst of the 2-mouse groups either with CellROX or with an extracellular superoxide detecting method like cytochrome C or Diogenes.

6. PLOS authors have the option to publish the peer review history of their article (what does this mean?). If published, this will include your full peer review and any attached files.

Reviewer #1: No

Reviewer #2: No

---

## [Author Response · Author response to Decision Letter 0]

19 Jan 2022

Reviewer 1

In the present study “Organ dysfunction and outcomes in a mouse model of acute surgical sepsis are independent of resistin concentration”, the authors compared the resistin deficient mice and resistin transgenic mice in CLP model. Here are some major concerns to be addressed: 

1. To confirm the function of human resistin in mice, the authors need to detect a classical signal of resistin in the transgenic mice in comparison to the background knockout mice. 

We thank the reviewer for this comment and we hope to provide some additional clarity. Physiologic effects of human resistin in this transgenic mouse model have been demonstrated and published by Lazar et al (Diabetes, 2011), from whom we have obtained the mouse model used in these experiments. We hypothesized that acute sepsis (in contrast to sham surgery) causes a dramatic increase in blood resistin concentration that recapitulates what happens in acutely septic patient. Our hypothesis was confirmed in the results demonstrated in Fig. 1. We then hypothesized that elevated blood resistin concentrations would also correlate with (although not necessarily cause) organ dysfunction, as it does in humans. This was one of the hypotheses to be tested, rather than an a priori assumption of the study.

2. Judging from the survival experiment, almost 25% of the sham operation transgenic mice reached euthanasia point. The percentage should be very low. The surgery itself may have introduced too much incomparable variation and the current research is not valid. 

We agree with the reviewer that the percentage appears high, although we believe that this death was an outlier in a group with a low sample size (n=3 for Retn+ sham-operated). We clarify this point by the addition of a sentence in the legend for Figure 5. Our conclusion regarding the robustness of our sham operation is supported by the fact that none of the 5 sham-operated Retn- mice died prior to the censoring point. 

3. The resistin transgenic mice have better acute neutrophil response, better subacute macrophage response and better neutrophil function. 24 hours after sepsis induction, these mice stopped inflammatory reaction. The resistin deficient mice showed more disorganized immune response. However, the transgenic mice do not show better bacterial clearance or survival. The authors need to explain why. 

Although we do agree that flow cytometry data indicates a different inflammatory blood cell profile at 6h and 24h with resistin production, these data appear less consistent in peritoneal fluid. We have delved deeper into the interplay between outcomes and cellular profiles in pages 20-21 of the manuscript (Discussion section). Our discussion also raises the possibility of model artifact, given the lack of a difference in bacterial clearance or survival as the reviewer points out.

4. The authors need to explain why resistin transgenic mice have low MHC-II expression. 

Please see reply to comment#3. 

5. The authors need to show basic metabolic conditions of the mice. Do resistin transgenic mice have lower body weight, less food intake, lower metabolism rate, or more bacterial growth in the cecum compared to resistin deficient mice? 

Metabolic data for both mouse strains has been previously published (Lazar et al., Diabetes 2011, in which it was the primary focus of the paper). We have made reference to this paper on page 5 of the revised manuscript. We have added baseline weights of the mice used in our experiments to the Results section (‘General Characteristics of Septic Mice’ subsection of the manuscript)

6. The authors need to show sham-operational groups in figure 3. 

Figure 3 has been updated with the requested data.

7. Macrophages should not be in circulation. The authors need to correct interpretation of flow cytometry result. 

We thank the reviewer for this correction, and we have revised the terminology (and flow cytometry gating) accordingly. Please note that, due to the need to re-gate peritoneal macrophages and monocytes some of our results and conclusions have changed, as noted in the text corresponding to Fig. 3.

8. The current study did not find any direct evidence of organ dysfunction. Also, this study compared high concentration of resistin versus almost no resistin. This is not a comparison between different concentration of resistin. The title should not include “organ dysfunction” or “concentration”.

We thank the reviewer for these valid points. We have revised the manuscript title, accordingly, to “Resistin production does not affect outcomes in a mouse model of acute surgical sepsis.”

Reviewer 2

In this manuscript Bonavia and his coworkers explored the in vivo physiological significance of the human resistin on sepsis severity and organ dysfunction using the cecal ligation and puncture surgical sepsis model on resistin knockout (Retn-/-) and transgenic mice expressing the human resistin gene and its regulatory elements on Retn-/- background. Since in the literature controversial data were published whether resistin is just a biomarker of sepsis severity and organ failure or it could be a therapeutic target, this manuscript has important findings that aims to clarify the in vivo clinical significance of resistin.

As for the formal aspects, the manuscript is logical, well-written and readable, however, there are some typos that should be corrected in the final paper: for instance, in the first sentence of the Results section mice and rats were both mentioned as experimental subjects, even though only mice were used in the present manuscript.

We thank the reviewer for pointing this out, and we have made the relevant correction.

Although the goals are clear and the conclusions are in line with the presented Figures, there are some issues that needs to be addressed:

1. Authors should elaborate in the Introduction for a sentence why was it necessary to express the human resistin gene in resistin KO mice? Why they couldn`t simple use resistin WT and resistin KO mice for this study since they mention that the sequence homology between the human and murine gene is ~ 60% and similar transcriptional regulation is presumed. According to NCBI Gene the human and murine Retn have completely different expression pattern.

We thank the reviewer for this comment and for the opportunity to provide additional clarity. The differing expression pattern is precisely the reason why we chose to eliminate the murine resistin gene and replace it with the human counterpart and its upstream transcriptional elements. The seminal article: ‘The genomic organization of mouse resistin reveals major differences from the human resistin: functional implications” (Ghosh et al., Gene, 2003) delves into the significant genomic and transcriptomic differences between mouse and human resistin. The salient points are that “while at the mRNA level the human resistin shows 64.4% sequence identity with its mouse counterpart, the mouse resistin genomic sequence displays only 46.7% sequence identity with the human resistin and is almost three times bigger than the human resistin.” Significant differences also exist in intronic sequences and protein binding sites, which likely account for its completely different cellular expression pattern. The model developed by Lazar et al (Diabetes 2011) and used in our experiments recapitulate the desired expression pattern (ie, that observed in humans). Equally important is the fact that resistin expression in mouse adipocytes is suppressed by inflammatory cytokines (eg. TNFa) whereas human resistin expression in macrophages is induced by TNFa. This difference is critical since one of the goals of our investigations was to develop a translationally-relevant model of sepsis-induced resistin expression. Suppression of (mouse) resistin by pro-inflammatory cytokines in acute sepsis would not allow us to investigate its effects on organ dysfunction in this setting. We have added more information to the ‘Introduction’ section to clarify this salient point (page 3 of manuscript text).

2. Since the human variant of Retn expressed primarily in macrophages, was this well-established mouse model Retn-/-/+ verified previously that it has similar expression pattern?

Yes that is correct. This was done and published by Lazar et al (Diabetes, 2011), who is the creator of this transgenic mouse model and from whom our lab has obtained the mice strains used for our experiments. 

3. What can be known about the other 3 mouse resistin-like molecules? Can any of those contribute to the effects shown in these experiments? If not, why?

This is an excellent question, and the subject of a recent review by Pine et al (Cytokine, 2018). However, given that (1) our primary interest was to elucidate the role of human resistin in sepsis, and (2) our experiments did not include the relevant controls to make any inferences about the role of mouse resistin in sepsis, we purposefully steered away from speculation in our manuscript without adequately powered, supporting data. Since mouse resistin appears to be suppressed in the context of acute inflammation, however, we would be skeptical about it having a role in acute sepsis.

4. The authors detected significant differences in LPS and PMA stimulated peritoneal neutrophil ROS production measured by CellROX Deep Red reagent. Neutrophils were stimulated for 2 hours than incubated with CellROX for 30 min and finally a cumulative value was measured. Authors should measure and show kinetics of oxidative burst of the 2-mouse groups either with CellROX or with an extracellular superoxide detecting method like cytochrome C or Diogenes.

Please see the data in Supplentary Figure S3 for fluorescence emission kinetics within 65min of addition of CellROX Deep Red reagent. Data is derived at 10 second intervals, starting 5min following addition of CellROX Deep Reagent, until 65min following the addition of CellROX. Neutrophils were taken from two animals from each group (Retn+ and Rko). Relative fluorescent values (y axis) represent fluorescent units minus control at that time point. It demonstrates that there is no ‘peak’ emission prior to or after the 30min time point at which we report sample fluorescence in our manuscript.

---

## [Decision Letter · Decision Letter 1]

28 Feb 2022

Resistin production does not affect outcomes in a mouse model of acute surgical sepsis

PONE-D-21-27883R1

Dear Dr. Bonavia,

We’re pleased to inform you that your manuscript has been judged scientifically suitable for publication and will be formally accepted for publication once it meets all outstanding technical requirements.

Kind regards,

Partha Mukhopadhyay, Ph.D.

Section Editor

PLOS ONE

Additional Editor Comments (optional):

Reviewers' comments:

Reviewer's Responses to Questions

**Comments to the Author**

1. If the authors have adequately addressed your comments raised in a previous round of review and you feel that this manuscript is now acceptable for publication, you may indicate that here to bypass the “Comments to the Author” section, enter your conflict of interest statement in the “Confidential to Editor” section, and submit your "Accept" recommendation.

Reviewer #1: All comments have been addressed

Reviewer #2: All comments have been addressed

2. Is the manuscript technically sound, and do the data support the conclusions?

Reviewer #1: (No Response)

Reviewer #2: Yes

3. Has the statistical analysis been performed appropriately and rigorously? 

Reviewer #1: (No Response)

Reviewer #2: Yes

4. Have the authors made all data underlying the findings in their manuscript fully available?

Reviewer #1: (No Response)

Reviewer #2: Yes

5. Is the manuscript presented in an intelligible fashion and written in standard English?

Reviewer #1: (No Response)

Reviewer #2: Yes

6. Review Comments to the Author

Reviewer #1: (No Response)

Reviewer #2: The authors have answered all my raised questions and I have no further comments. I recommend accepting the manuscript for publication.

7. PLOS authors have the option to publish the peer review history of their article (what does this mean?). If published, this will include your full peer review and any attached files.

Reviewer #1: No

Reviewer #2: No

---

## [Editor Report · Acceptance letter]

4 Mar 2022

PONE-D-21-27883R1 

Resistin production does not affect outcomes in a mouse model of acute surgical sepsis 

Dear Dr. Bonavia:

I'm pleased to inform you that your manuscript has been deemed suitable for publication in PLOS ONE. Congratulations! Your manuscript is now with our production department. 

Kind regards, 

on behalf of

Dr. Partha Mukhopadhyay 

Section Editor

PLOS ONE